# Rapid Uniformity Analysis of Fully Printed SWCNT-Based Thin Film Transistor Arrays via Roll-to-Roll Gravure Process

**DOI:** 10.3390/nano13030590

**Published:** 2023-02-01

**Authors:** Yunhyok Choi, Younsu Jung, Reem Song, Jinhwa Park, Sajjan Parajuli, Sagar Shrestha, Gyoujin Cho, Byung-Sung Kim

**Affiliations:** 1Department of Semiconductor Systems Engineering, College of Information and Communication Engineering, Sungkyunkwan University, Suwon-si 16419, Republic of Korea; 2Department of Biophysics, Institute of Quantum Biophysics, Research Engineering Center for R2R-Printed Flexible Computer, Sungkyunkwan University, Suwon-si 16419, Republic of Korea; 3Department of Intelligent Precision Healthcare Convergence, Sungkyunkwan University, Suwon-si 16419, Republic of Korea

**Keywords:** roll-to-roll gravure, thin film transistor arrays, uniformity, probing instrument

## Abstract

The roll-to-roll (R2R) gravure process has the potential for manufacturing single-wall carbon nanotubes (SWCNT)-based thin film transistor (TFT) arrays on a flexible plastic substrate. A significant hurdle toward the commercialization of the R2R-printed SWCNT-TFT array is the lack of a suitable, simple, and rapid method for measuring the uniformity of printed products. We developed a probing instrument for characterizing R2R gravure printed TFT, named PICR2R-TFT, for rapidly characterizing R2R-printed SWCNT-TFT array that can present a geographical distribution profile to pinpoint the failed devices in an SWCNT-TFT array. Using the newly developed PICR2R-TFT instrument, the current–voltage characteristics of the fabricated SWCNT-TFT devices could be correlated to various R2R-printing process parameters, such as channel length, roll printing length, and printing speed. Thus, by introducing a characterization tool that is reliable and fast, one can quickly optimize the R2R gravure printing conditions to enhance product uniformity, thereby maximizing the yield of printed SWCNT-TFT arrays.

## 1. Introduction

Maintaining a product’s performance uniformity is one of the most critical elements in the manufacturing of commercial products. Efforts to provide product uniformity always come along at a high cost. However, ensuring the product’s high quality provides more economic benefits overall. In Si-based semiconductor industries, manufacturers invest heavily in developing testing protocols at multiple stages to ensure the final product’s uniformity. A good example of a midstream manufacturing test is the electrical die sorting process which can inspect and sort out defective devices on a single wafer, depending on the type and purpose of the devices. Since each manufactured product will be used as a component for an even more expensive product, sorting out defective devices before the final packaging process will provide a significant economic advantage to the manufacturer and the customer. In the field of printed electronics, the roll-to-roll (R2R) gravure process has been regarded as an inexpensive and sustainable manufacturing alternative for highly flexible and disposable electronic devices, such as sensors [1], biochips [2], and radio frequency identification tags [3]. As a good example of R2R gravure printed flexible electronics, single-wall carbon nanotube-based thin film transistors (SWCNT-TFTs) have been attracted by many research groups because of specific metrics such as high carrier mobility, excellent chemical stability, and mechanical flexibility [4,5,6,7,8]. In the R2R gravure printed SWCNT-TFTs, however, the formation of a uniform layer is one of the critical factors in achieving the printed devices that are uniform over TFT arrays [9,10,11,12]. Thus, developing a simple and rapid method for monitoring the devices’ uniformity of the R2R gravure printed products would be valuable and efficient to elucidate the influence of each printed layer of the SWCNT-TFTs while the other printed layers remain unchanged.

In this work, we explored a high throughput testing method to characterize and monitor the uniformity of the R2R gravure printed devices. We devised a probing instrument for characterizing the R2R gravure printed TFT array, called PICR2R-TFT, to measure printed devices’ current–voltage (I–V) characteristics. To demonstrate the functionality of PICR2R-TFT, we fabricated a 26 × 38 array of SWCNT-TFT on a 250 mm width roll of polyethylene terephthalate (PET) using two different printing processes: silver nanoparticle-based conducting ink and BaTiO_3_-based dielectric ink, each of which is used to print gate, drain-source, and dielectric both printed at the same speed. The SWCNT-based semiconducting ink, on the other hand, was printed under various printing speeds and channel lengths to figure out how the performance of the TFT devices was affected by the printed SWCNT layers. By analyzing the printed 26 × 38 array of SWCNT-TFT using the PICR2R-TFT, we demonstrated that the R2R gravure printed SWCNT layers can be optimized quickly and accurately to increase device yield.

## 2. Materials and Methods

An R2R gravure printing system (i-PEN, Hwaseong-si, Korea) with two printing units (Appendix A) was utilized to fabricate an SWCNT-TFT array on a PET substrate (250 mm wide, 100 µm thick, AH71D, SKC, Seoul, Korea). The overall R2R gravure printing conditions, including ink conditions (formulation), were identical to the conditions reported in our previous work [13,14]. The gate layer and dielectric layer were printed sequentially using Ag nanoparticle-based and BaTiO_3_-based inks. After printing on the PET film, all of the electronic inks were cured in an oven at 150 °C for 5 to 10 s, depending on the printing speed. After that, the SWCNT-based ink was printed on the dielectric/gate layers to form the active layer. The SWCNT-based ink formulation was identical to the ink reported in our previous work [14]. Lastly, the drain and source layers were overlay-printed on the SWCNT semiconducting layer using the same Ag nanoparticle-based ink. All printing processes were executed in a 23 ± 2 °C and 40 ± 5% humidity environment. The basic rheological electronic properties of the fabricated devices and other details can be found in our previously reported work [14].

The surface tension and viscosity of the electronic inks were measured under an ambient condition using a SV-10 Vibro viscometer (AND Co, Tokyo, Japan). To characterize and test the SWCNT-TFT array sample, a semiconductor parameter analyzer (Keithley 4200, Solon, OH, USA and Agilent, 4156C, Santa Clara, CA, USA) and LCR meter (4284 A, Hewlett Packard, Palo Alto, CA, USA) were used.

A semi-automatic array probing system (PICR2R-TFT) to rapidly characterize an R2R-printed SWCNT-TFT array was designed to characterize the R2R-printed SWCNT-TFT array, as shown in Figure 1a. It consists of four parts: base, fastening cover, probing instrument, and probe head with pogo-pins, as shown in Figure 2a,b. The array had 26 × 38 SWCNT-TFTs, where each TFT had two 0.5 mm × 0.5 mm pads for the drain and source layers. All TFTs in the same row shared one gate pad (Figure 1b). The distance between the drain pad and the source pad was 1.55 mm, and the distance between TFTs was 2.55 mm. The PICR2R-TFT can simultaneously characterize an entire row of the printed 26 × 38 SWCNT-TFT array.

Base: Any non-conductive solid material can be suitable for the base. We made the base plate for this system using Teflon, and the plate dimensions were 165.35 mm × 149.50 mm × 3.00 mm. To facilitate the maintenance of the PICR2R-TFT, we incorporated a groove into the back side of the plate.

Fastening cover: The fastening cover was custom-made using an anodized aluminum alloy (AL6050) with dimensions identical to the base plate: 165.35 mm × 149.50 mm × 3.00 mm. Proper alignment of the probing head is critical to consistently locating the identical position on each individual TFT in the array. Since the precise alignment of the instrument is critical, a novel alignment method was devised. This issue was solved by gluing a transparent film containing a blueprint of the TFT array on the base plate where the TFT array is required to rest (Figure 3a). The SWCNT-TFT array is held in place by pressure from the fastening cover, which engages the base plate through four screws. The main probing instrument is located at the appointed position by engaging the fastening cover. The probe head and the cover are designed with jigsaw-shaped teeth (Figure 3b inset) on each end that can interlock for perfect probing alignment. As soon as the handles of the probe head instrument are released, the probing head locates naturally at a correct probing position (Figure 3b). Next, the gate and drain voltages with reference to the source are applied for the precise I-V measurement system (Figure 3c).

Probing instrument (PICR2R-TFT): The probing instrument utilized multiplexers (ADG5206 and ADG5208 manufactured by Analog Devices Inc., Wilinton, MA, USA), as shown in Figure 4 and Appendix A (detailed circuit layout), to sequentially measure the TFT devices on the same row of the TFT array using a semiconductor parameter analyzer (4156C). Due to the very low current levels from the printed SWCNT-TFT, it is important to control the leakage current of the instrument itself. To confirm the reliability of the probing instrument, we measured the resistance and leakage current between two adjacent probes (Appendix A). The leakage current measurement was conducted by floating the instrument, and the probe-to-probe resistance was measured by probing a gold-plated metal plate. The measured leakage current was less than 150 pA and was considered suitable for measuring the I-V data of the printed SWCNT-TFT array. Furthermore, with the maximum ON current of about 1 μA flowing between the drain and source of the printed SWCNT-TFT, the voltage drop between the two ports was expected to be less than 0.5 mV. Therefore, we confirmed that the dedicated probing instrument was suitable for measuring the I–V characteristics of the printed SWCNT-TFT array.

Probe head with pogo-pins: Since the SWCNT-TFT array is printed on a flexible PET substrate, the conventional needle probes would penetrate it. Thus, we had to find an appropriate probe type that would not damage the printed SWCNT-TFT array substrate. Pogo-pins (Leeno Industrial Inc., Busan, Republic of Korea), spring-loaded pins, were selected for this purpose. Since the pogo-pin probes were located at specifically marked pads on the probe head holder (Appendix A), damaged probes can be easily identified and replaced.

PICR2R-TFT integrated control program: Selecting transmission protocols and constructing control algorithms are of importance and care to simultaneously control our instrument and the 4156C analyzer. The final program was written in Visual Basic .NET 6.0 language for execution on a Virtual Instrument Software Architecture Application Programming Interface (API). The probing instrument was controlled with serial transmission (RS-232), easily implemented in a .NET program. The integrated program instructs the instrument’s Micro-Controller Unit (MCU) to open the correct switch. The MCU (STM32F303RET6 manufactured by STMicroelectronics, Geneva, Switzerland) governs all multiplexers on the PCB. The key map of control flows is shown as a block diagram in Figure 4.

## 3. Results and Discussion

A complete inspection for fabrication uniformity of millions of printed TFTs is required to estimate the performance uniformity of the printed electronic devices on a large-scale array [1]. However, one of the challenges that have prevented the broad adoption of R2R-printed electronics in the commercial electronics industry was the lack of an automated measurement for evaluating the entire sheet of printed devices in a short period of time. Therefore, we introduce a quick probing method for characterizing the uniformity of R2R-printed devices. To develop a system that can measure some of the TFT parameters with speed and precision, we adopted and modified a probe card of the conventional equipment used to test devices mounted on ceramic-based wafers. As an electrical interface between automated test equipment (ATE) and devices on a wafer, a probe card can provide an identical test circumstance to thousands of devices on a wafer. Inspired by the probe cards widely used in the ceramic-based semiconductor industry, we designed an instrument that can efficiently distribute the semiconductor parameter analyzer’s resources while maintaining identical probing conditions. As a result, we developed the PICR2R-TFT probing system that can quickly characterize a large volume of R2R-printed SWCNT-TFT arrays.

To demonstrate the effectiveness of our newly designed PICR2R-TFT for characterizing a fully R2R-printed 26 × 38 SWCNT-TFT array, sample arrays were printed on a PET roll using three different electronic inks: Ag nanoparticle-based, BaTiO_3_ nanoparticle-based, and SWCNT-based inks (Figure 5a). The gate, dielectric, active, and drain-source layers were respectively printed through the R2R gravure printing process. The web tension and pressure of the impression rollers were maintained with accuracies of ±0.3 kgf and ±0.38 psi, respectively [15]. The control system for web tension and roll impression pressure from our previously reported setup was used without modification [15,16]. To prevent device failure from printing position errors, we incorporated a control system based on three charge-coupled-device (CCD) cameras into our R2R gravure printing system [13]. This control system allowed us to achieve a printing registration accuracy of ±25 µm in the machining direction (MD) and ±50 µm in the transverse direction (TD). Additionally, our R2R-printing unit was designed to actively compensate for the expansion of PET film as it passed through the 150 °C of drying chamber [17]. The gate, dielectric, and drain-source layers were continuously printed by employing two printing units with a web width of 250 mm and the same printing speed of 90 mm/s. To ensure proper ink transfer, the cell structures in the R2R gravure cylinder were customized to account for the viscosity and surface tension of the specific electronic inks’ properties based on our previously reported method [14]. After printing the dielectric layer on the gate layer, the web was rewound to print the subsequent layers. This R2R gravure printing sequence was repeated until the 26 × 38 SWCNT-TFT array was completely fabricated. Next, the SWCNT semiconducting active layer was printed on the dielectric layer with various printing speeds. Lastly, the drain and source layers were printed with a registration accuracy of ±50 µm in the MD on top of the active layer. Figure 5b presents a photographic image of the roll of R2R-printed 26 × 38 SWCNT-TFT arrays with insets showing an array constructed with 520 µm channel widths and different channel lengths (15, 25, and 60 µm).

To the printed 26 × 38 SWCNT-TFT array, the base and fastening cover were positioned and fastened, as shown in Figure 3a. The PICR2R-TFT probe head was mounted on this assembly (Figure 3b), and the gate and drain voltages were then applied to the SWCNT-TFT array from an external power supply system (Figure 3c). After completing one measurement set for a single gate line (38 SWCNT-TFTs) in 30 s, the probe was manually moved to the following gate line. The PICR2R-TFT device was designed to be able to measure every single gate line on the array for 30 s and was capable of handling multiple measurement setups simultaneously. Once the printed 26 × 38 SWCNT-TFT array was loaded onto the PICR2R-TFT system, the typical transfer (Figure 6a) and output (Figure 6b) curves of a single TFT in the R2R-printed SWCNT-TFT array, and all 26 lines of 38 SWCNT-TFTs in the array were characterized (Figure 6c). Due to the use of the BaTiO_3_ nanoparticle (piezo material)-based dielectric ink and the small size of each SWCNT-TFT, the R2R-printed SWCNT-TFTs exhibited slightly larger hysteresis than those presented in our previous study [14].

From the capacitance–voltage (C-V) measurements [15], we confirmed a dielectric layer capacitance of 13 nF/cm^2^, slightly higher than in our previous report. The small structural dimensions of the TFTs caused a coffee ring effect on the printed dielectric layers. When we characterized the 988 TFTs from the 26 × 38 SWCNT-TFT array constructed with channel lengths of 60 µm (Figure 6d–g), the device yield was 51% with an average threshold voltage of 15.0 ± 5.3 V, average mobility of 0.017 ± 0.02 cm^2^/Vs, an average on–off current ratio of 21.6 ± 12, and an average transconductance of 9.48 ± 3.68 µS. The mobility (µ_FE_) was calculated from maximum transconductance according to the equation: µ_FE_ = 2L(g_m_)WCi [1], where Ci and g_m_ are the gate capacitance per unit and transconductance, respectively. Due to the rough, smaller gate width (520 µm) and a thin dielectric layer (1–1.2 µm thick), the attained device mobility was lower than previously reported [13].

Since the transfer of SWCNT ink during the R2R gravure printing process of the active layer is a key factor that influences the printed device’s characteristics [1], we studied the role of printed SWCNT layer by implementing three different printing speeds using the newly developed PICR2R-TFT system. To clearly see the role of the SWCNT layer in the R2R-printed 26 × 38 SWCNT-TFT array, the ink transfer mechanism in R2R gravure printing should be understood through the fluid dynamics of the capillary bridge of the liquid ink since the plastic substrate, and engraved cells of the gravure cylinder are contacted and then, detached [18] with employed ink between them. Thus, with a high-viscosity ink with high web transfer speed, the ink transfer phenomena will be easier to study since the ink transfer ratio depends mainly on the capillary number (Ca), which increases as increasing the ink transfer ratio. Therefore, the high-viscosity ink was usually employed with the high web transfer speed to provide appropriate ink transfer in the R2R gravure printing process. However, unlike the high viscosity (900–1500 cP) of Ag nanoparticle-based ink, the semiconducting SWCNT ink in this R2R gravure process has a low viscosity of 20–30 cP at room temperature. In some cases, the viscosity of SWCNT-based ink can be increased up to 100 cP to print SWCNT-TFTs. However, the performance of TFTs with high viscosities of SWCNT ink will be poorer than low-viscosity ones because of the interference of large amounts of binders and additives in ink. Although organic molecule-based semiconducting ink can be used to print a semiconducting layer of TFTs, a few cases can be seen in the literature regarding fabricating devices using the R2R-printing process [19] due to the degradation problem in organic semiconductors. Thus, we printed the 26 × 38 SWCNT-TFT arrays as a reference sample through the R2R gravure system to study the role of the semiconducting layer under three different printing speeds (30, 60, and 90 mm/s) with different channel lengths and different printing lengths (Figure 7, Figure 8, Figure 9 and Figure 10), while the other layers were all R2R gravure printed using a constant printing speed of 90 mm/s. We observed a linear relationship between the web transfer speed for printing SWCNT layers and successful device yield, as shown in Figure 7a–c. When the array was printed at a speed of 30 mm/s, only one TFT sample out of the 988 units in the 26 × 38 SWCNT-TFT array could be classified as a “pass” (Figure 7a). Similarly, we observed poor device yields (0–4.8%) in both cases (15 µm and 25 µm of channel length-based TFTs) at a speed of 30 mm/s (Appendix A), proving the critical role of the printing SWCNT as showing a large variation in yields of the printed device. The shortened channel length case (15 µm) showed better device yield (~4.8%) at a speed of 30 mm/s. It may be due to the easy formation of SWCNT networks to connect the channel between the source and drain layers. As the printing speed increased to 60 mm/s and 90 mm/s, the device yield increased to 29.8% and 51.0%, respectively, as shown in Figure 7b,c. Interestingly, at a printing speed of 60 mm/s, the passing devices (working devices) were primarily observed near the front end of the array. This could be explained by poor SWCNT ink transfer from the roller and the downward directional flow of SWCNT ink during the drying process.

The device yield for TFTs with different channel lengths (15, 25, and 60 µm) with the same channel width (520 µm) ranged from 14.8% to 51.0%, as shown in Figure 8a–c. The lowest device yield (14.8%) was observed for channel lengths of 25 µm, while the highest device yield (51.0%) was obtained for channel lengths of 60 µm. The higher device yield was observed in the longer channel length because the overlay printing registration accuracy of the R2R gravure printing system was in the range of MD (machine direction) ± 50 µm and TD (transverse direction) ± 25 µm. As the channel length was increased, the effect of the overlay printing registration accuracy (MD ± 50 µm and TD ± 25 µm) on the printing of drain-source electrodes became less significant.

Acceptable device yields (47.0–51.0%) could be obtained for the printed SWCNT-TFT arrays by the R2R gravure printing system where 60 µm channel length and 520 µm channel width at a printing speed of 90 mm/s were employed up to 6 m long, as shown in Figure 9a–c. Figure 9a–c showed the “pass” and “fail” diagrams for selected 26 × 38 SWCNT-TFT arrays along the printed 6 m roll. Near the end of the printing length (~6 m), the device yield dropped to as low as 29.7% due to SWCNT ink agglomeration losing the ink stability, which caused open characteristics in the 26 × 38 SWCNT-TFT arrays as the printing time increased.

After analyzing the collected data by using the PICR2R-TFT system, we can conclude the linear relationship between device yield and printing speed of SWCNT ink with high repeatability and reproducibility (Figure 10a) because a similar tendency was observed in our PICR2R-TFT system for each printing condition. Moreover, the structural design of the TFT can be studied by checking the behaviors of all R2R-printed SWCNT-TFTs since the device yields were quickly obtained from various channel lengths between 15 and 60 µm using the PICR2R-TFT system (Figure 10b) in less than 3 h. If we characterized all samples manually, it would take more than 24 h. Thus, the PICR2R-TFT system will be a powerful tool employed to quickly characterize a couple of thousands of all R2R-printed TFTs under various printing conditions so that the optimization process to reach the practical device yields will be much easier than that through manually characterizing the devices.

## 4. Conclusions

In this study, we developed a fast characterization tool, the PICR2R-TFT system, that can analyze an R2R-printed SWCNT-TFT array (26 × 38 = 988 TFTs). To demonstrate the capabilities of the PICR2R-TFT system, 26 × 38 SWCNT-TFT arrays were fabricated entirely using the R2R gravure printing process. Then, we were able to test the printed TFT array using the PICR2R-TFT system that extracts device characteristics to rapidly analyze printing and optimizing parameters by changing various conditions, such as printing speed, channel length, and printing roll length. Furthermore, utilizing the PICR2R-TFT system, the key parameters in ink transfer, printing length, and TFT’s structural dimensions were quickly reviewed and modified for good yields of samples. It was observed that the choice and the quality of SWCNT inks in the R2R gravure printing process played an important role in terms of device quality and yield. Based on the experiments using a 26 × 38 SWCNT-TFT array, the highest device yields were realized at approximately 51% from the TFT array with channel lengths of 60 µm and a printing speed of 90 mm/s. On the other hand, a very limited number of working SWCNT-TFT samples were obtained at a printing speed of 30 mm/s for all channel lengths (12, 25, and 60 µm) due to SWCNT ink transferring and wetting issues at the low printing speed. Since the proposed PICR2R-TFT system is able to completely characterize all printed samples quickly, it is possible to avoid any misleading result caused by partial inspection when figuring out the optimal printing conditions. In addition, the PICR2R-TFT system can give one guidance that is statistically data-driven, not randomly selected sample-based. It is also believed that the PICR2R-TFT system can be expandable to evaluate the various R2R-printed devices, such as integrated logic gates as well as TFT-active-matrix array. Although the structure of the probing instrument may need to be altered to analyze the integrated logic gates, such as printed processors, the fundamental operating concept and algorithm can be applied with ease to test the various printed circuit and systems without significant changes.

Despite the PICR2R-TFT system’s ability to extract parameters much faster than manual measurements, considerable time is still required for the massive data analysis. Thus, implementing machine learning in the PICR2R-TFT system is expected to be a promising solution for fast characterization and statistical analysis of collected data. By using the PICR2R-TFT system, failed devices that do not meet thresholds of electrical performances can be identified and analyzed along with their printing parameters, which can train a machine learning algorithm to achieve optimal TFT structures and printing conditions. In this work, we developed and demonstrated the PICR2R-TFT system that has the potential as a standard tool for the rapid characterization and statistical analysis of millions of R2R-printed devices.

## Figures and Tables

**Figure 1 nanomaterials-13-00590-f001:**
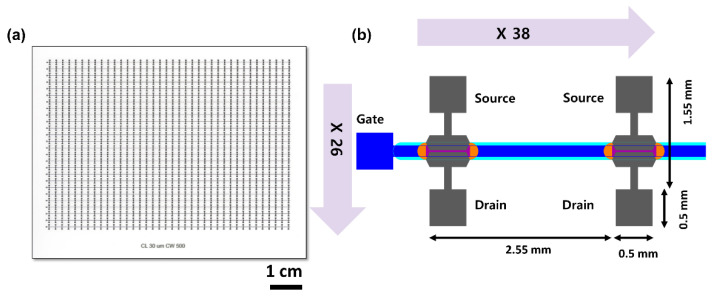
(**a**) Diagram of the 26 × 38 SWCNT-TFT array. (**b**) Enlarged view of a single TFT in the array.

**Figure 2 nanomaterials-13-00590-f002:**
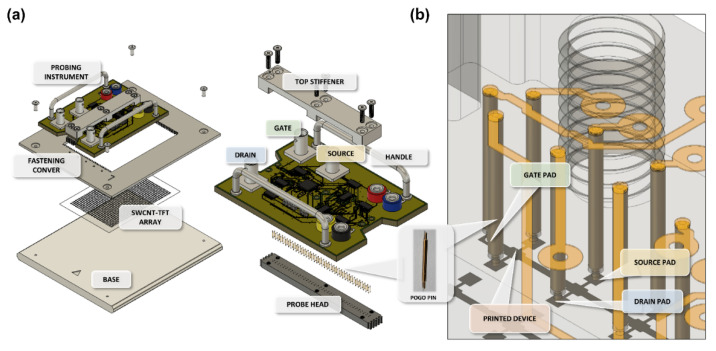
(**a**) Assembly of the PICR2R-TFT probing system. (**b**) Enlarged image for the pin contact structure.

**Figure 3 nanomaterials-13-00590-f003:**
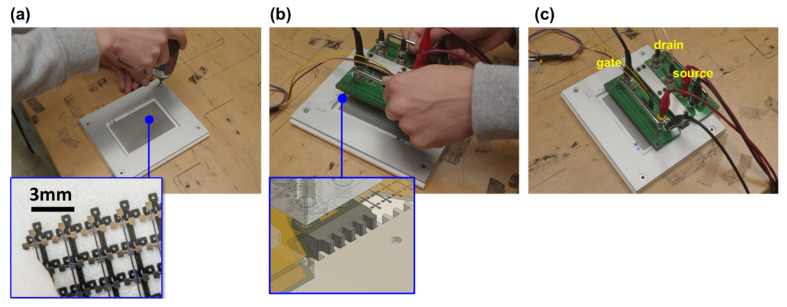
Images depicting the stepwise process for assembling the PICR2R-TFT array probing system: (**a**) Alignment of the SWCNT-TFT array onto the base and installation of the cover to hold the array in place. (**b**) Fitting the probe instrument onto the fastening cover with an inset showing the jigsaw-shaped teeth to properly align the probe head. (**c**) The connection of gate, source, and drain lines in the PICR2R-TFT system.

**Figure 4 nanomaterials-13-00590-f004:**
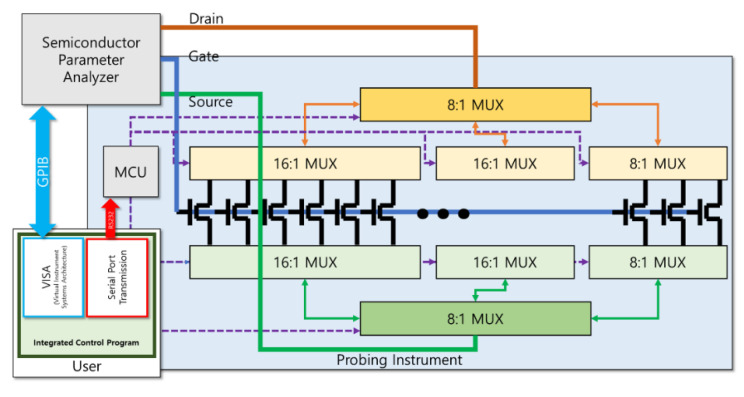
The key map of controlling flow: the integrated program in a PC controls the semiconductor parameter analyzer and the probing instrument simultaneously. After selecting a switch for measurement by commanding the MCU of the probing instrument, the integrated program orders the semiconductor parameter analyzer to sweep gate voltages and receives the measured drain current data.

**Figure 5 nanomaterials-13-00590-f005:**
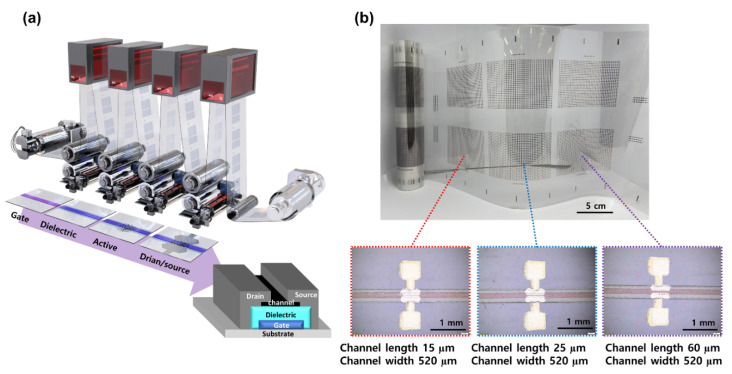
(**a**) Artistic rendering of the R2R gravure printing process for fabricating a 26 × 38 SWCNT-TFT array. (**b**) Roll image of R2R gravure printed 26 × 38 SWCNT-TFT arrays and the enlarged optical images of SWCNT-TFT with three different channel lengths (15, 25, and 60 µm).

**Figure 6 nanomaterials-13-00590-f006:**
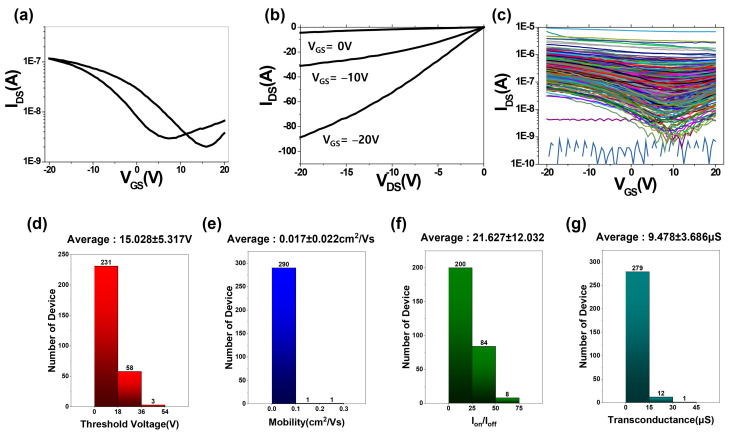
Typical characteristics of a single TFT from the 26 × 38 SWNCT-TFT array: (**a**) Transfer and (**b**) output characteristics of single SWCNT-TFT. (**c**) Overlayed transfer characteristics for all 26 × 38 SWNCT-TFTs in the array. (**d**) Calculated threshold voltage, (**e**) mobility, (**f**) on–off current ratio, and (**g**) transconductance of all 26 × 38 SWNCT-TFTs in the array.

**Figure 7 nanomaterials-13-00590-f007:**
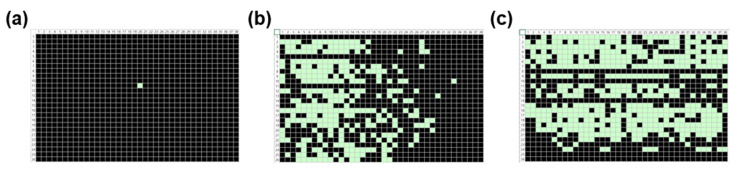
Statistical maps generated from PICR2R-TFT measurements of the 26 × 38 SWCNT-TFT arrays with a channel length of 60 μm printed at various speeds: (**a**) 30 mm/s, (**b**) 60 mm/s, and (**c**) 90 mm/s (light box: “pass”, dark box: “fail”).

**Figure 8 nanomaterials-13-00590-f008:**
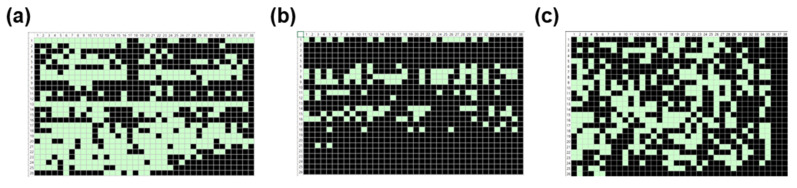
Statistical maps generated from PICR2R-TFT measurements of the 26 × 38 SWCNT-TFT arrays with a printing speed of 90 mm/s containing different channel lengths: (**a**) 15 μm, (**b**) 25 μm, and (**c**) 60 μm (light box: “pass”, dark box: “fail”).

**Figure 9 nanomaterials-13-00590-f009:**
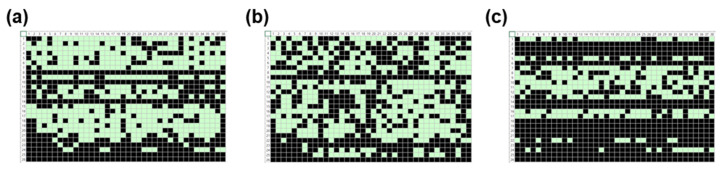
Statistical maps generated from PICR2R-TFT measurements of the 26 × 38 SWCNT-TFT arrays with a printing speed of 90 mm/s containing channel lengths of 60 μm at various lengths of a 6 m printing run: (**a**) 0 m, (**b**) 3 m, and (**c**) 6 m (light box: “pass”, dark box: “fail”).

**Figure 10 nanomaterials-13-00590-f010:**
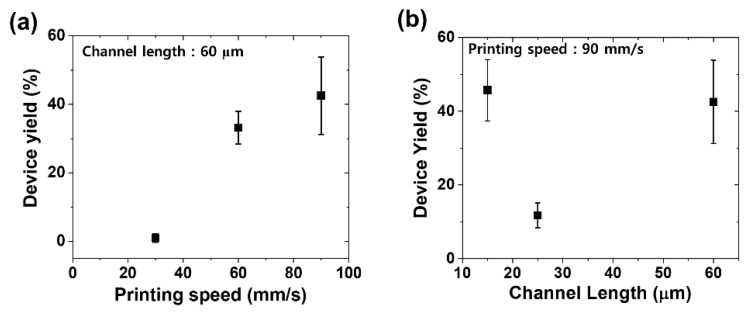
Relationship between device yield and printing conditions: (**a**) device yield depending on different printing speeds and (**b**) device yield depending on channel lengths.

## Data Availability

Data can be obtained on request to gcho1004@skku.edu or bskimice@skku.edu. They may also be accessed from the Appendix A after publication of the manuscript.

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
