# Peer review of "Rapid Uniformity Analysis of Fully Printed SWCNT-Based Thin Film Transistor Arrays via Roll-to-Roll Gravure Process"

_nanomaterials, 2023, doi:10.3390/nano13030590_

Round 1

Reviewer 1 Report

The key focus of this paper is on the development of a rapid characterization tool, PICR2R-TFT system, to analyze SWCNT-TFT array. The topic is good and the analyses are reasonable. I recommend the acceptance of the manuscript after some minor revisions.

1) In general, the grammar used in this paper can be improved. I suggest that some of the professional English language authors, or an editor, take a more active role in the re-writing and editing of the manuscript.

2) The conclusion is a little short. The authors could briefly state what are the shortcomings of their analyzing method.

3) Thin film formation is quite critical for various applications, such as transistors, sensors, and thermoelectrics. However, to generate uniform thin films, many factors are actually counted, such as additives, fillers, crystallinity, and preparation methods. The following papers are recommended to be cited as they study how to produce uniform thin film from different angles. This will provide the readers with a bigger picture of how the additives including CNT affect the formation of thin film. The papers include:

1) Transferrable, Large‐area and Highly conductive PEDOT: PSS Film and its Application for Flexible Thermoelectric Generator. This article focuses on the preparation methods and how it can be used in applications.

2) High-performance PEDOT: PSS-based thermoelectric composites. This article studies how carbon-based nanomaterials, like CNT and others, affect the formation of thin film. It also provides the mechanism studies and the influence of carbon nanomaterials affecting thin film performance.

3) Self-Organization of PEDOT: PSS Induced by Green and Water-Soluble Organic Molecules. This article studied how additives’ crystallinity influences the thin film formation and the corresponding performance.

4) Gallium-Doped Zinc Oxide Nanostructures for Tunable Transparent Thermoelectric Films. This article studies how another type of inorganic material affects the formation and performance of thin film.

Reviewer 2 Report

Comment for nanomaterials-2177012 is listed as follows,

(1)  In the Keywords, please check the "roll-to-roll gravure" and "thin film transistor arrays", they didn't been used in the paragraph of manuscripts.

(2)  In the Abstract, please change the "a probing instrument characterizing R2R-printed TFT array (PICR2R-TFT)" into the "a probing instrument characterizing R2R-printed TFT (PICR2R-TFT) array".

(3)  In the section 1. Introduction, line 35, please change the "the R2R gravure" into the "the roll-to-roll (R2R) gravure"; line 41, also please change the "a roll-to-roll (R2R) gravure " into the "a R2R gravure". Lines 43-44, please change the "system, ,PICR2R-TFT" into the "system, probing instrument characterizing R2R-printed TFT (PICR2R-TFT)".

(4)  In page 4, line 120, please check "Figure S2"; line 140, please check "(Figure S3)".

Reviewer 3 Report

This is an interesting work; however, before proceeding to the next step, the authors should address the following comments.

1. The language of the manuscript has to be improved.

2. Provide a more in-depth discussion of related previous works.

3. Authors should also provide more meaningful discussions regarding the repeatability and reproducibility of the conducted tests/analysis.

4. In the “Conclusion” section, I recommend presenting more quantitative data as the main results of the research study.
